# Study of the Microstructure and Crack Evolution Behavior of Al-5Fe-1.5Er Alloy

**DOI:** 10.3390/ma12010172

**Published:** 2019-01-07

**Authors:** Ming Li, Zhiming Shi, Xiufeng Wu, Huhe Wang, Yubao Liu

**Affiliations:** College of Materials Science and Engineering, Inner Mongolia University of Technology, Hohhot 010051, China; shizm@imut.edu.cn (Z.S.); 20171800197@imut.edu.cn (X.W.); Huhe2009@imut.edu.cn (H.W.); liubobo2008@163.com (Y.L.)

**Keywords:** Al-5Fe-Er alloy, microstructure, in situ tension, crack evolution

## Abstract

In this work, the microstructure of Al-5Fe-1.5Er alloy was characterized and analyzed using X-ray diffraction (XRD), scanning electron microscopy (SEM), transmission electron microscopy (TEM) and energy-dispersive X-ray spectroscopy (EDS) techniques. The effect of microstructure on the behavior of crack initiation and propagation was investigated using in situ tensile testing. The results showed that when 1.5 wt.% Er was added in the Al-5Fe alloy, the microstructure consisted of α-Al matrix, Al_3_Fe, Al_4_Er, and Al_3_Fe + Al_4_Er eutectic phases. The twin structure of Al_3_Fe phase was observed, and the twin plane was {001}. Moreover, a continuous concave and convex interface structure of Al_4_Er was observed. Furthermore, Al_3_Fe was in the form of a sheet with a clear gap inside. In situ tensile tests of the alloy at room temperature showed that the crack initiation mainly occurred in the Al_3_Fe phase, and that the crack propagation modes included intergranular and trans-granular expansions. The crack trans-granular expansion was due to the strong binding between Al_4_Er phases and surrounding organization, whereas the continuous concave and convex interface structure of Al_4_Er provided a significant meshing effect on the matrix and the eutectic structure.

## 1. Introduction

Being among the lightest structural materials, aluminum alloy has the advantages of low density, high specific strength and stiffness, good thermal conductivity and excellent electromagnetic shielding and anti-radiation properties, due to which, it has applications in various fields, including manufacturing, aerospace and electronic communication [1,2,3,4]. Although 8000 series aluminum alloy is the most widely used casting aluminum alloy in industrial production, its poor mechanical properties at room temperature are one of the bottlenecks for further development and application. The alloying effect is one of the most important means to improve the mechanical properties of aluminum alloy. The existing research [5,6,7,8,9,10,11,12,13] has shown that erbium (Er) can improve the mechanical properties of aluminum alloys. After adding Er, a stable structure of rare earth aluminum compounds is formed in the alloy, which can inhibit the precipitation of Al_3_Fe phase. At present, researchers have conducted a lot of experimental work to improve the microstructure and mechanical properties of 8000 series aluminum alloy using Er. Karnesky et al. [8,9,10,11,12,13,14] found that the microstructure of 8000 series aluminum alloy was refined due to the addition of Er (0–0.8 wt.%) and formed new granular or needle-like Al_4_Er compounds. The study of Che et al. [15,16,17,18,19] suggested that, after the addition of Er in the proportion of 0.3–1.0 wt.%, the precipitation of Al_4_Er phase at grain boundaries enhanced the mechanical properties of 8000 series aluminum alloy. Dorin et al. [20,21,22,23,24,25,26] reported the formation of Al_4_Er phase in 7073 after the addition of 1 wt.% Er, whereas both the tensile and yield strengths of the alloy improved after alloying.

At present, there are only a handful studies involving in situ observations of the crack propagation behavior in 8000 series aluminum alloy. Based on previous experiments, this paper adopted in situ dynamic observation using scanning electron microscopy (SEM) to study the crack initiation and crack propagation behavior of Al-5Fe-1.5Er alloy at room temperature. The paper also discussed the influence and mechanism of microstructures on the evolution of crack.

## 2. Test Methods and Procedures

Pure aluminum, Al-15%Fe and Al-10%Er intermediate alloys were used as raw materials to prepare Al-5Fe-1.5Er alloy using vacuum induction melting furnace (SKG-0005, S.Y. Furnace Works, Shanghai, China). During the smelting process, the alloy was kept in vacuum, protected by argon, and then, shaped using metal mold casting. The HCS-140 high-frequency infrared ray carbon sulfur analyzer (SHIMADZU labX XRD-6100, Kyoto, Japan) was used to analyze the composition of alloy. S-3400N scanning electron microscope (HITACHI, Tokyo, Japan) and its corresponding energy spectrometer were used to observe the microstructure and fracture appearance in the test alloy. The composition and the content of elements in each phase were also analyzed. X-ray diffraction (XRD) was used to conduct the phase analysis using D/MAX-2500/PC type X-ray diffractometer, which was procured from Rigaku Corporation (Tokyo, Japan). The transmission electron microscopy (TEM) samples of the alloy were prepared using Gatan 691 precision polishing system. The JEM2010 transmission electron microscope (JEOL, Tokyo, Japan) and its corollary energy-dispersive spectrometer were used to conduct the microstructure analysis and energy-dispersive spectrum analysis (EDS) of the testing alloy. 

The specimen size for in situ tensile tests, conducted at room temperature, is shown in Figure 1. After mechanical grinding and electrolytic polishing on the prefabricated gap of alloy, the alloy was put to in situ tensile tests on the tensile stage under FEI Quanta 650 scanning electron microscope (FEI, Lausanne, Switzerland) in a vacuum environment. The scanning rate was set to be 0.5 mm/min.

## 3. Results and Discussion

### 3.1. XRD and SEM Analyses 

The XRD results of the tested alloy are shown in Figure 2a. Based on the PDF card analysis (PDF card is 47-1420,17-0666,46-1028), the main phase compositions were detected, which included α-Al, Al_3_Fe, Al_4_Er and Al_10_Fe_2_Er. Figure 2b shows the picture of the tested alloy’s SEM image. The results for the EDS analysis of the selected points are presented in Table 1. It can be seen that the morphology of iron phase in the alloying tissue changed significantly due to the addition (1.5 wt.%) of rare earth elements. Point E represents the Al matrix, whereas the results showed that a small amount of Fe element was dissolved in the matrix. The main alloying elements of Point B were Al, Er, and Fe. When the influence of Fe element’s content is disregarded, Point B is regarded as η-Al_4_Er phase. The particle phase at Point A is mainly composed of Al and Fe elements, and is regarded as the θ-Al_3_Fe phase. The white needle-like phase at Point C is mainly composed of Al, Fe and Er elements. With regards to the results of XRD analysis, the phases were confirmed to be: Al_10_Fe_2_Er. The main alloying elements of Point D were Al, Er and is regarged as η-Al_4_Er phase. Further analysis of the SEM results found that Al_4_Er and its peripheral α-Al formed a eutectic structure in the alloy. Moreover, part of the white needle-like phase (Al_10_Fe_2_Er) at the grain boundaries attached itself to η-Al_4_Er phase. Furthermore, part of the needle-like phase precipitated inwardly onto the matrix.

### 3.2. TEM Analysis

The Al_3_Fe phase formed the bottom-center of the monoclinic structure, whereas its lattice constants were found to be: *a* = 1.549 nm, *b* = 0.808 nm, *c* = 1.247 nm and *β* = 107.72°. The Al_4_Er phase belongs to cubic system and Pm3m space group, which lattice constants were 0.4215 nm. The TEM picture of the Al_3_Fe phase of the tested alloy and the results of the selected area electron diffraction pattern (SAED) are shown in Figure 3. Figure 3a shows the morphology of the second phase, which was obtained using electrolyzation. In addition, the second-phase θ-Al_3_Fe and η-Al_4_Er phase are shown in Figure 3b. It can be seen that both the θ-Al_3_Fe and η-Al_4_Er combine to grow together. The SAED results of θ-Al_3_Fe and η-Al_4_Er phases are shown in Figure 3c and Figure 3d, respectively. After calibration for the two sets of SAED patterns, the η phase was found to be Al_4_Er, and had the zone axis orientation of [130] (Figure 3c). Additionally, the θ phase was found to be Al_3_Fe, and had the zone axis orientation of $\left[\bar{1}02\right]$ (Figure 3d). This result further verifies the SEM and EDS results of the white needle-like phase, which is the eutectic phase consisting of both the η and θ phases.

Figure 3b indicates that a mass of plane defects penetrated the whole θ-phase grain along the growth direction. According to the SAED pattern shown in Figure 3b, the plane defects were considered to be twin crystal and stacking fault [27,28,29,30]. The twin crystal appeared in the (001) plane, and a mass of stacking fault in θ-phase led to the appearance of streaks in the SAED pattern. Figure 4a shows the high-resolution transmission electron microscopy (HRTEM) of the twin crystal. High-density regular twinning was evident in the θ-crystal. The HRTEM image showed a complex atomic arrangement, while the twinning was difficult to distinguish, as shown in Figure 4b. However, the adjacent atomic arrangements were slightly different along the (001) plane, thereby confirming the atomic arrangements of twinning seen in Figure 4b. The formation of (100) and (201) twinning may require more energy than (001) twinning. The nucleus contained only (001) twinning and stacking faults, whereas the other twinning routes were suppressed. The (110) and (110)_T_ planes formed by (001) twinning offered a mass of steps favoring atomic stacking along the (001) plane, which led to the continuous growth of (001). Notably, the twinning nucleation may appear and grow in the growing θ-crystal, whereas the (001) twinning remained the sole mechanism.

The existence of a twin structure in Al_3_Fe may result from high structural stress in alloy. Metal casting has a relatively fast cooling rate. When the alloy is cooled to room temperature, the structural stress in the alloy is in an imbalanced state and has a tendency to be released spontaneously. When the stress exceeds the critical shear stress of Al_3_Fe phase, the twin structure Al_3_Fe phase is induced to form.

η-Al_4_Er phase has a cubic structure, and its lattice constants were found to be: *a* = 0.4215 nm, α = *β* = *γ* = 90°. Figure 5 shows the TEM, EDS and SEM results of Al_4_Er phase. Figure 5a shows the SEM results of needle-like phase at grain boundary. According to the EDS results of needle-like phase shown in Figure 5b,d, the proportions of Al and Er elements in the outer edge of the phase was relatively large. The calibration of SAED on the needle-like phase showed that it was the Al_4_Er phase, and had [130] as the zone axis orientation (Figure 3c). This result is consistent with the SEM and EDS results of needle-like phase. Further investigation revealed that the grain boundaries of the needle-like Al_4_Er phase were coated on the outside edge of Al_3_Fe. It was concluded that Al_10_Fe_2_Er was a composite phase, in which the outer layer was Al_4_Er, while the inner was Al_3_Fe. This composite phase can be identified as the θ-Al_3_Fe/η-Al_4_Er phase. The end face of the needle-like Al_4_Er was parallel to the bonding interface of the α-Al matrix along the length direction, whereas it was intermittent along the width direction, thus exhibiting an obvious concave convex appearance. At this moment, Al_4_Er started the adsorption growth along the width direction, which means that all the atoms on the concave convex interface will combine into the crystal. Therefore, the crystal, even under small under-cooling, can grow up quickly. However, the uneven degree of diffusion of Al and Er atoms on the concave convex interface of Al_4_Er front edge resulted in the formation of wall-like Al_4_Er. The change in the interfacial morphology of Al_4_Er showed that the needle-like Al_4_Er phase tended to grow into short rod-like shapes. Figure 5c shows the SEM image of the needle-like Al_4_Er phase and Al_3_Fe phase inside the grain. It can be seen that the two phases grow perpendicular to each other, so that the dense Al_4_Er phase hinders the diffusion of iron and inhibits the growth of Al_3_Fe. The shape of smooth-edged needle-like Al_4_Er phase was in good agreement with the results for Al matrix. The η-Al_4_Er phase exhibited evident fibrous characteristics, as observed from the TEM images of the two zone axes (Figure 5e,f). The average distance of fibers was approximately 20 nm. The η-phase should be disordered during the generation of the alloy phase because of the attached growth. The η-Al_4_Er phase was the dominant reaction product and performed an important role in controlling the generation of alloy phase in molten aluminum. In addition, the finger-like morphology of η-Al_4_Er was attributed to the distinctive crystal structure. The orthorhombic structure consisted of 30% vacancies along the *c*-axis [001], which offered a rapid diffusion path. However, an η-phase attached growth means that all the atomic positions in the lattice were the same. None of the 30% of vacancies along the c-axis [001] were offered. Furthermore, there were many dislocation lines inside the brittle Al4Er phase.

### 3.3. Analysis of the Results of In Situ Tensile Tests at Room Temperature

Figure 6 shows the in situ tensile testing of the alloy at room temperature. Figure 6a shows the relationship between the load and displacement of the alloy specimens. The observation of in situ tension started from the position of the arrow in the diagram. The tensile scan images of different corresponding positions are shown in Figure 6b–o. Figure 6b shows the back-scattered electron scanning images of the specimen for the external load of 200 N. Meanwhile, the specimen was in the elastic deformation stage, whereas for the position of the sample reserved gap front, no crack source was formatted. When the load was continuously increased (Figure 6c), micro cracks began to form in the gaps of the specimen due to stress concentration. No significant change was observed in the middle part of the specimen. A close magnification of the selected area shown in Figure 6c (see Figure 6d) clearly shows that a large number of secondary phases (needle-like θ-Al_3_Fe + η-Al_4_Er, particulate Al_3_Fe phase, and island-like Al_4_Er) were dispersed in the α-Al matrix of the alloy specimen. Moreover, the microstructure of the alloy was dense. Figure 6d shows the elliptic Regions A and B (eutectic Al_3_Fe+ Al_4_Er phase) as a comparative observation area. When the external load was further increased (Figure 6e), the cracks in the gap grew. Due to the effect of stress concentration, some Al_3_Fe phase at the front of the crack’s source (magnified area in Figure 6e) and in the middle of the specimen began to generate fine cracks. For the time being, no obvious plastic deformation was observed in the central part of the specimen (Figure 6f and a selected area of Figure 6e), and no micro-cracks formed in the aluminum compound. When the external force was further increased, the main crack expanded rapidly (Figure 6g). There were obvious slip bands and secondary cracks on both sides of the main crack. The formation of secondary cracks is a result of further expansion of micro-cracks in the Al_3_Fe/Al4Er phase. When the selected areas in Figure 6g are enlarged (Figure 6h), one can see the widened micro-cracks formed by the broken island-like Al_4_Er phase on both sides of the main crack. The needle-like Al_3_Fe/Al_4_Er phase and the granular Al_3_Fe phase were broken. Al_3_Fe/Al_4_Er cracked several times along the length direction. The width of micro-cracks formed by the broken rare earth aluminide rupture was relatively small, and the bonding between the rare earth aluminum compounds and the interface of the matrix was good, which indicates that the needle-like Al_3_Fe/Al_4_Er phase and particulate Al_4_Er phase in the alloy can effectively resist changes in external load. In addition, the movement of the slip line was hindered by the rare earth aluminide, which further improved the tensile strength of the alloy. With the continuation in tension, the front part of the main crack front showed a Z-shaped expansion (seen from the selected area in Figure 6i in the selected block area), whereas next to the front of main crack, an obvious plastic deformation region (slip band formation) appeared in the matrix. When the main crack expanded to the area, the crack path extended forward along the slip plane (steps); thus, forming the front end of “Z”-shaped crack. This kind of winding crack propagation can improve the fatigue strength of alloy [12]. The Z-shaped crack continued to expand forward. When the crack passed the Grains 1 and 2, one can see from the path of the main crack that there were two forms of crack expansion. One was the trans-granular, while the other was the intergranular crack expansion. The micro-cracks near the main crack grain were connected to each other, and expanded a pathway for their extension. When the main crack expanded to Grain 3, it did not extend forward along the grain boundary, though the trans-granular went through the grain’s interior instability region (see the area surrounded by blue curve in Figure 6j). The red arrow in the diagram points to the direction of crack propagation. The particulate Al_3_Fe phase in the coarse θ + η eutectic structure at the grain boundary (the area selected by yellow lines) was broken. Additionally, the weak bonding between the eutectic structure and the matrix’s interface led to cracks and the subsequent separation of the two. Further observation revealed that the formation of a needle-like Al_3_Fe/Al_4_Er phase and particulate Al_4_Er phase formed at the grain boundary increased the connection between the eutectic structure and matrix, especially the concave and convex interface structure of Al_4_Er increased its meshing area with the surrounding tissue, which further hindered the separation between primary α-Al crystal and Al element in the eutectic structure. The step is to block the Al from the α-Al and the eutectic structure. Therefore, the interface needs to provide a larger external load to initiate and propagate cracks, which results in trans-granular cracking of the main crack. When the external force was further increased, the stress concentration area was in the front of the main crack, making the surface of the sample unstable (see the selected Regions C and D in Figure 6k). There are slip bands of different orientations in the region. Further observation on the fracture behavior of the crack through the selected Regions C and D (Figure 6m,n) found that each piece of Al_3_Fe/Al_4_Er and matrix interface bonded closely and slid along the direction of deformation for some distance, which showed that the bonding strength between the needle-like Al_3_Fe/Al_4_Er and the matrix interface hindered the slip of the matrix, forcing the need-like phase to break into several segments of almost the same length (about 2 μm) along the length direction. 

The fracture was mainly caused by stress concentration, which was borne by the convex and concave interface of Al_4_Er. The force of the stress concentration was stronger than the bond strength between the Al_3_Fe/Al_4_Er eutectic structure and the matrix. The crack shown in Figure 6n continued moving forward and went through the unstable region, causing the brittle Al_3_Fe phase to break and separate. When the tensile strength was increased, the alloy specimen fractured quickly. The surface of specimen fracture (Figure 6o) along the crack direction was non-homogeneous. Further observation of the selected region in the fracture (Figure 6p) showed a black hole at the bottom right, which should have been caused by the fractured Al_3_Fe phase. The nearby residual Al_3_Fe was exposed at the surface of the fracture matrix on the right side of the main crack, and fractured along the direction of stress, which showed that the main crack produced secondary micro-cracks going through the large sized Al_3_Fe_._ Under the action of external force, the secondary micro-cracks extended to the nearby Al_3_Fe phase, releasing the stress concentration, which was produced by Al_3_Fe and the surrounding matrix. The micro-cracks continued to propagate to the matrix. Under the effect of sheer stress, the matrix formed into a slippage step and converged to form dimples at the edge of matrix slippage step through the extension of micro-cracks of Al_3_Fe_2_, whereas, the effect of the cross-slip led to the formation of a tear ridge at the fracture surface.

Figure 7 shows the schematic of crack propagation path. Since the Al_3_Fe at the grain boundary can easily initiate micro-cracks, the Al_3_Fe/Al_4_Er eutectic structure broke when the main crack passed. The bonding between Al_3_Fe and matrix interface was weak, which caused detachment of the two, making the cracks extend along the grain boundaries. When the rare earth aluminide precipitated from the eutectic structure at the grain boundaries, the compounds exhibited a better effect on the meshing between the eutectic structure and the matrix, and improved the mechanical stability of the eutectic structure at grain boundaries. This resulted in the formation of an unstable region inside the grains, and produced the expansion of the trans-granular crack.

The EDS analysis of the selected point from the fracture surface (Figure 8b) showed that the point consisted of Al, Fe, indicating that it was on the grain boundary. Therefore, the fracture mode of the alloy was a combination of intergranular fracture and quasi-cleavage fracture.

## 4. Conclusions

(1) The main phase composition of Al-5Fe-1.5Er alloy included α-Al matrix, Al_4_Er, eutectic Al_10_Fe_2_Er phase and Al_3_Fe phase. There was a twin structure in Al_3_Fe, while the twin plane was the {001}. Al_4_Er had a concave and convex interface structure. Al_10_Fe_2_Er was a composite phase, in which, the outer layer was Al_4_Er, and the inner was Al_3_Fe. The composite phase was referred to as θ-Al_3_Fe/η-Al_4_Er.

(2) The in situ tensile test at room temperature showed that the cracks were mainly formed inside the coarse particle Al_3_Fe. The mode of crack propagation included the crack growth mode, which consisted of intergranular expansion and intergranular propagation. Intergranular propagation was the result of strong binding ability between the phase distributed along the grain boundaries (Al_4_Er) and the surrounding interface. The pinning effect was remarkable. The concave and convex interface structure of Al_4_Er had a strong meshing effect on the eutectic structure and the matrix, which improved the mechanical stability between the eutectic structure at grain boundaries and the interface of the matrix.

## Figures and Tables

**Figure 1 materials-12-00172-f001:**
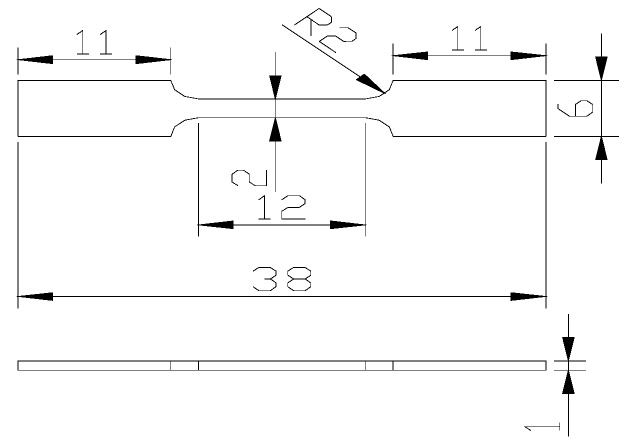
Geometry and size of tensile specimen (unit: mm).

**Figure 2 materials-12-00172-f002:**
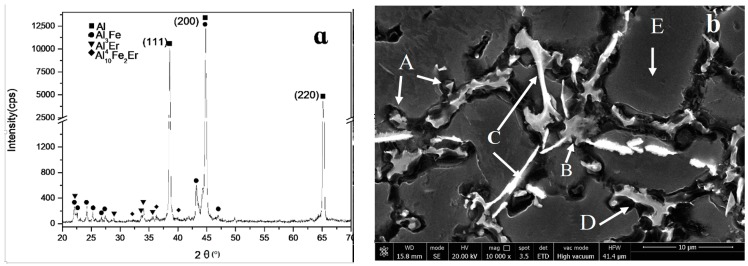
X-ray diffraction (XRD) analysis (**a**) and scanning electron microscopy (SEM) image (**b**) of the as-cast alloy.

**Figure 3 materials-12-00172-f003:**
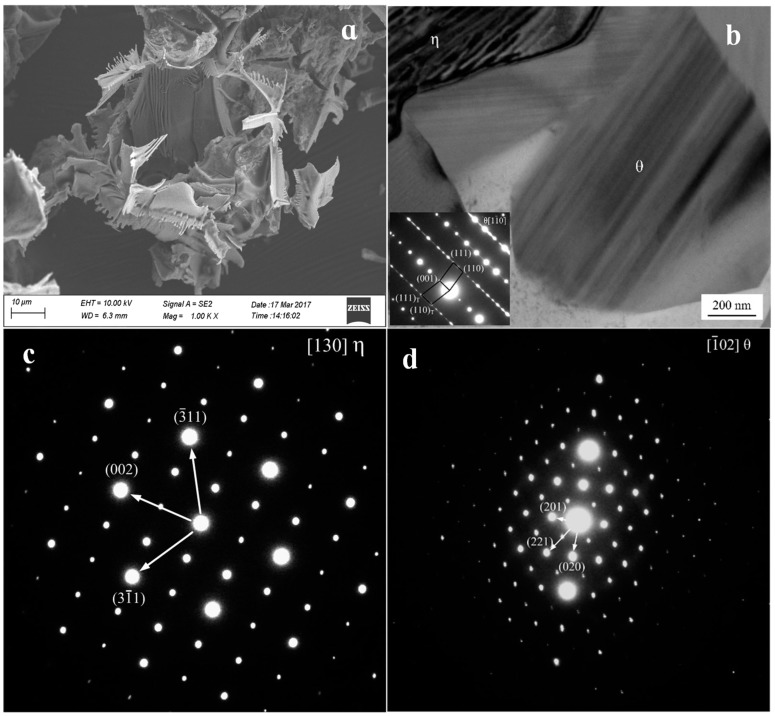
SEM image of the second-phase (**a**) and partial enlargement map (**b**) selected area electron diffraction (SAED) pattern of the Al_4_Er in [130] zone axis (**c**) and the Al_3_Fe in $\left[\bar{1}02\right]$ zone axis (**d**).

**Figure 4 materials-12-00172-f004:**
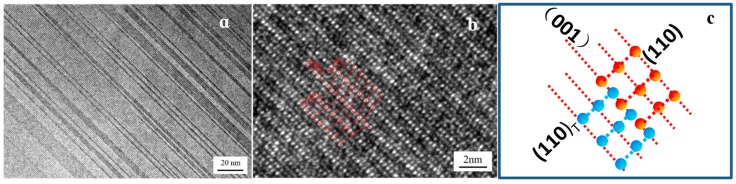
High-density twin morphology (**a**), high-resolution transmission electron microscopy (HRTEM) image of (001) twinning of Al_3_Fe phase (**b,c**).

**Figure 5 materials-12-00172-f005:**
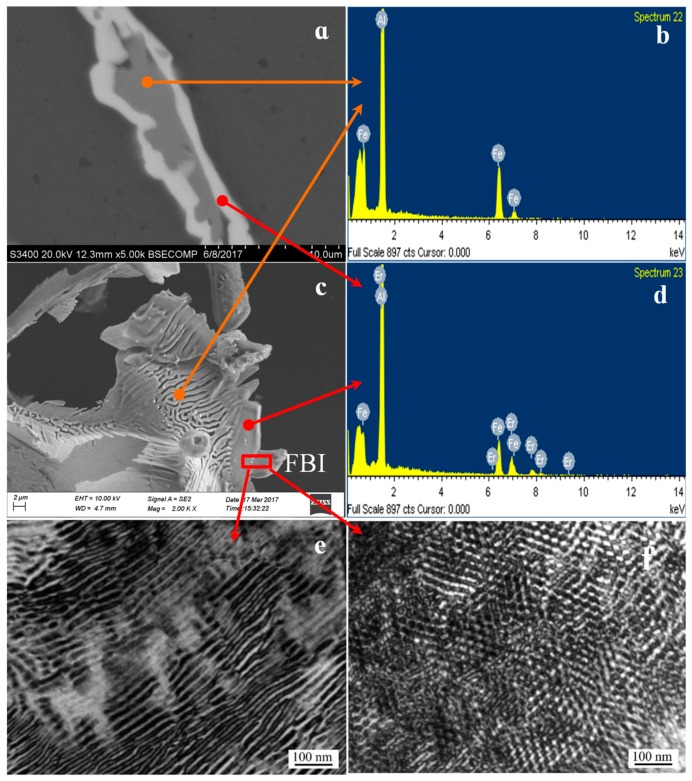
Transmission electron microscopy (TEM) images, SEM and EDS of Al_4_Er phase. (**a,c**) SEM image of Al_4_Er in grain boundary; (**b,d**) EDS analysis; (**e,f**) TEM image of Al_4_Er in grain.

**Figure 6 materials-12-00172-f006:**
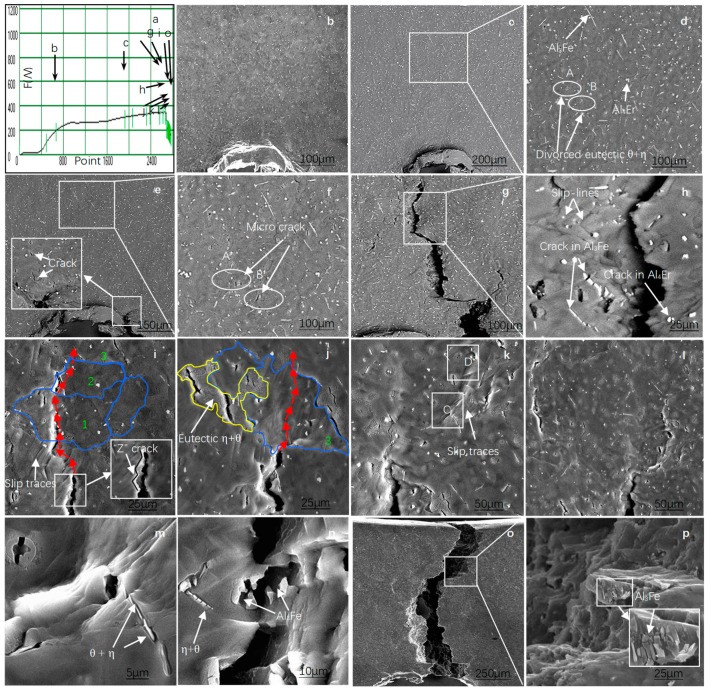
In situ tension test of the alloy at room temperature.(**a**) Load-position curve; (**b**) SEM image of Position b shown in (**a**); (**c**) SEM of Position c shown in (**a**); (**d**) Enlarged image of the selected area shown in (**c**); (**e**) SEM image at F = 340 N; (**f**) Enlarged image of the selected area shown in (**e**); (**g**,**h,i,j,k,l**) SEM images of Positions g, h, i, j, k, and l shown in (**a**); (**m**) Enlarged image of the selected area C shown in (**k**); (**n**) Enlarged image of the selected area D in (**k**); (**o**) SEM image of the fractured specimen; (**p**) Enlarged image of the selected area in (**o**).

**Figure 7 materials-12-00172-f007:**
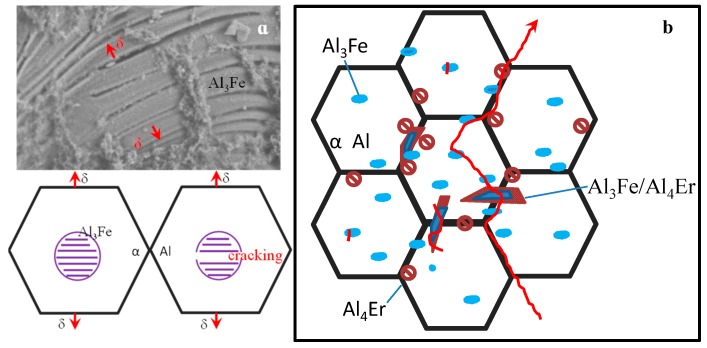
Schematic of the crack propagation in Al_3_Fe (**a**) and in alloy (**b**).

**Figure 8 materials-12-00172-f008:**
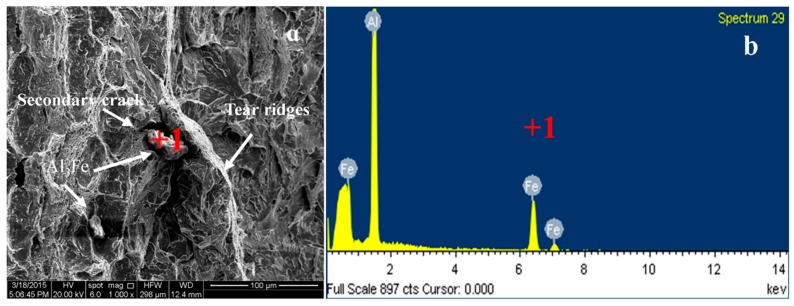
SEM image (**a**) and EDS analysis (**b**) of the alloy.

**Table 1 materials-12-00172-t001:** Energy-dispersive X-ray spectroscopy (EDS) analysis of the phases in the tested alloy.

Element (at.%)	Al	Fe	Er
A	62.66	37.34	-
B	82.38	0.64	16.98
C	65.94	15.14	18.91
D	81.96	-	18.04
E	100	-	-

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
