# Peer review of "Study of the Microstructure and Crack Evolution Behavior of Al-5Fe-1.5Er Alloy"

_materials, 2019, doi:10.3390/ma12010172_

Reviewer 1 Report

The paper deals with the analysis of an Al-5Fe-1.5Er alloy and is divided into two sections. In the first part the authors describe the microstructure of the Al-5Fe-1.5Er alloy. In the second part the damage evolution in a single tensile test performed in a SEM is described. The paper contains a variety of experimental findings, but unfortunately in some cases the relationship is not recognizable.

The text contains a lot of spelling, punctuation and grammatical mistakes and therefore the paper is very hard to read. The sum of necessary corrections exceeds an acceptable level, therefore the reviewer recommends to reject the paper and suggest the authors to carefully revise the paper and submit a revised paper with with a content reduced to the essential points.

Author Response

This paper analyzes the effect of rare earth Er on the second phase in Al-5Fe-1.5Er aluminum alloy. The addition of rare earth Er can change the precipitation process of the second phase. Al3Er precipitates first, and the precipitated Al3Er hinders the precipitation process and growth process of Al3Fe, forming Al3Fe and the inclusion of Al3Er, which affects the distribution of iron-rich phase. The increase of the second phase and the change of the distribution law will all affect the crack behavior of the aluminum alloy. This is the main line of writing in this article, I look forward to receiving your testimony. The mistakes in English grammar and words have been revised. Please give your judgment in your busy schedule.

Reviewer 2 Report

The manuscript reports the influence of Er addition to  an Al-Fe alloy of the 8000 series. A modification of the phase composition is described and the role of newly formed particles in the course of an in-situ straining in SEM is discussed. The manuscript contains several interesting results including the description of crack formation and propagation. However, the way of the results presentation is  very difficult to understand, mainly due to English language and style used throughout the manuscript. It contains numerous mistyping errors and also several scientific inaccuracies and rather speculative conclusions not fully supported by the experiment.

The following comments might improve the quality of the manuscript.

23  Aluminum is not the lightest structural material

34 et al → et al. also further in the text

36  Al4Er … 4 is an index

55  Gatan691 → Gatan 691

60  Give units in captions - for example (in mm …)

67  give the explanation of PDF (powder diffraction file?)

72-81  Analysis in point D is not referred in the text

82  Tab 1: at.% could provide better information

85 β=107.72 o give dot instead of circle

85 provide crystallographic data about Al4Er phase already here

86-96 Fig 3a is not a TEM image but SEM image

88  Invalid formulation

89  two dots

94  indicate which phase (do you mean C in fig. 2?)

97 Fig 3c and d are SAEDs from Fig. 3b? or different particles

99 it is not clear from the experiment that planar defects are oriented in the growth direction

111 Fig.4b schematics in red could be read with big difficulties

114   FBI in Fig. 5c stands for what?, Provide better scale bar in Fig. 5a

125  The calibration of the selected … → The analysis of …

129-131  the composite Al3Fe/Al4Er probably could not give special reflections of Al10Fe2Er in XRD pattern.

147-151 reformulate this paragraph and give a proof about vacancies or a reference

152 – 241   The whole paragraph 2.3 is written in a manner which is difficult to be understand by the reader and should be rewritten.

244-246  No Er is present in the particle

248 Change the formulation of the sentence

References: use proper journal abbreviations

English must be improved, check for mistyping errors

Author Response

First of all, thank you very much for your professional and meticulous review. I admire your academic and scientific attitude. I have revised your various suggestions, please give your judgment in your busy schedule, and look forward to receiving your affirmation and your admirable professional review comments.

Reviewer 3 Report

The work carried out by the present authors is well arranged. i strongly believe that the quality of the manuscript is high enough to be accepted in Materials after a minor revision.

 1. The unit used on tensile sample figure should be shown. 

2. The english needs to be improved

Author Response

Thank you very much for your affirmation of this paper. The mistakes in English grammar and words have been revised. Please give your judgment in your busy schedule.

Round  2

Reviewer 1 Report

The paper, especially the readability has been significantly improved. Only some minor errors should be corrected. A greater Attention should be paid to the references section. A revision of this section ist strictly recommended.

Line 201:           αAl          replace by                  α-Al

Line 204:              main crack                         the main crack

Line 209:              the need-like phase          the needle-like phase

Line 222/223:     The surface of specimen fracture (Figure 6o) along the crack direction was non-homogeneous.                        Please revise this sentence (not understandable)

References:          (Line 267-296):   Please revise, several spaces, points and commas are missing, some points are double, some names and titles are written in capital letters. Please check also the abbreviations of the Journals (different abbreviations for the same Journal are used)

Author Response

The mistakes in English grammar and words have been revised. The references have been modified.Please give your judgment in your busy schedule.

Reviewer 2 Report

Included corrections increased significantly the quality of the manuscript. Small typing errors (see below) and extensive corrections of references should be performed.

201 – αAl → α-Al  

230 sheer → shear 

References should be carefully read and corrected. Non-standard abbreviations of journals names are used. The same journal is mentioned in capital letters, second time in a different form. Names of authors are written in a non-uniform manner.

p { margin-bottom: 0.25cm; line-height: 115%; }

Author Response

(The authors gave the same response as above.)
